# Phylogenomic analysis and *Mycobacterium tuberculosis* antibiotic resistance prediction by whole-genome sequencing from clinical isolates of Caldas, Colombia

**Lusayda Sánchez-Corrales**[1], **Olga Lucía Tovar-Aguirre**[2], **Narmer Fernando Galeano-Vanegas**[3,4], **Paula Alejandra Castaño Jiménez**[2], **Ruth Arali Martínez-Vega**[5], **Carlos Ernesto Maldonado-Londoño**[6], **Johan Sebastián Hernández-Botero**[7,8], **Fernando Siller-López**[2,9]*

1 Maestría en Investigación en Enfermedades Infecciosas, Universidad de Santander, Bucaramanga, Santander, Colombia, 2 Programa de Bacteriología, Universidad Católica de Manizales, Manizales, Caldas, Colombia, 3 Instituto de Investigación en Microbiología y Biotecnología Agroindustrial, Universidad Católica de Manizales, Manizales, Caldas, Colombia, 4 Departamento de Biotecnología, BIOS Centro de Bioinformática y Biología Computacional, Manizales, Caldas, Colombia, 5 Facultad de Salud, Universidad de Santander, Bucaramanga, Santander, Colombia, 6 Centro Nacional de Investigaciones de Café –Cenicafé, Manizales, Caldas, Colombia, 7 Grupo de Investigación Médica, Escuela de Medicina, Universidad de Manizales, Manizales, Caldas, Colombia, 8 Grupo de Resistencia Antibiótica de Manizales, Manizales, Caldas, Colombia, 9 Programa de Microbiología, Universidad Libre, Pereira, Risaralda, Colombia

* fsiller@ucm.edu.co

**Data Availability Statement:** The data sets analyzed in this study, with relevant

## Abstract

*Mycobacterium tuberculosis (M. tuberculosis)* was the pathogen responsible for the highest number of deaths from infectious diseases in the world, before the arrival of the COVID-19 pandemic. Whole genome sequencing (WGS) has contributed to the understanding of genetic diversity, the mechanisms involved in drug resistance and the transmission dynamics of this pathogen. The object of this study is to use WGS for the epidemiological and molecular characterization of *M. tuberculosis* clinical strains from Chinchiná, Caldas, a small town in Colombia with a high incidence of TB. Sputum samples were obtained during the first semester of 2020 from six patients and cultured in solid Löwenstein-Jensen medium. DNA extraction was obtained from positive culture samples and WGS was performed with the Illumina HiSeq 2500 platform for subsequent bioinformatic analysis. *M. tuberculosis* isolates were typified as Euro-American lineage 4 with a predominance of the Harlem and LAM sublineages. All samples were proven sensitive to antituberculosis drugs by genomic analysis, although no phenotype antimicrobial tests were performed on the samples, unreported mutations were identified that could require further analysis. The present study provides preliminary data for the construction of a genomic database line and the follow-up of lineages in this region.

epidemiological metadata, are available at the NCBI database as the Biosamples (SAMN20847272, SAMN20847273, SAMN20847274, SAMN20847275, SAMN20847276, SAMN20847277), covered on the Bioproject PRJNA755956.

**Funding:** Funding of our study were obtained from the Universidad Católica de Manizales (Grant 033-2019), Universidad de Manizales (Grant RES-088-16-11-2018) and Universidad de Santander (Postgraduate research fellowship to Lusayda Sánchez-Corrales). Authors who received salary from the Universidad Católica de Manizales were Olga L Tovar-Aguirre, Narmer F Galeano-Vanegas and Fernando Siller-López. Ruth A Martínez-Vega received her salary from the Universidad de Santander, and Johan S Hernández-Botero received his salary from the Universidad de Manizales. The funders had no role in study design, data collection and analysis, decision to publish, or preparation of the manuscript.

**Competing interests:** The authors have declared that no competing interests exist.

## Introduction

*M. tuberculosis (MTB)* is considered one of the most lethal pathogens, with a toll of 1 billion people suffering tuberculosis (TB)-related deaths in the last two centuries. The World Health Organization (WHO) in their annual report of 2019 depicted an estimate of 10 million new cases and 1.7 million deaths per year [1], making *MTB* the deadliest single infectious agent before the arrival of SARS-CoV-2. As in many developing countries, Colombia has presented a steady increase in tuberculosis incidence in the last six years from 23 to 27.6 cases per 100,000 habitants in 2019. In Caldas, a department located in the center of the country, TB is considered endemic causing 34 new cases and 2.31 deaths per 100,000 habitants at its peak in 2017, making this malady a primary public health concern in the region due to the progressive aging of the population, emerging immunosuppressive treatments, sociodemographic and comorbidities conditions. Chinchiná (47,929 habitants in 2012), a small town in the center of Caldas, is key to the region's coffee harvesting processes which lead to the arrival of hundreds of coffee pickers from all over the country at harvesting season. Endemic social issues like poverty and homelessness make a perfect combination for TB to thrive in Chinchiná resulting in one of the highest TB incidence rates in the country (71.5 cases per 100,000 habitants) [2]. WGS offers an opportunity for the molecular epidemiology surveillance for TB in the region. The development of local capacities bioinformatic analysis and the reduction on overall genome sequencing costs make whole genome sequencing (WGS) an increasingly accessible alternative for molecular epidemiologic studies [3]. With the genetic information obtained by WGS compared to other analysis, drug resistance profiles can be predicted for most of the current antibiotic treatment, enabling a wide monitoring of drug resistance.

The aim of this study is to explore the potential use of WGS in the context of routine surveillance, describing the epidemiological and molecular characteristics of circulating *M. tuberculosis* strains from Chinchiná, Caldas using minimal sequencing and computational resources.

## Materials and methods

### Population and sample

Patients suspected of pulmonary tuberculosis due to respiratory symptoms screened by routine surveillance in the first semester of 2020 were considered for the study. Demographic information of the patients was obtained from clinical records and epidemiological data from the National surveillance system (SIVIGILA) of Colombia. Seventeen cases of TB were notified from the health authorities of Chinchiná, Caldas to the SIVIGILA system. Six subjects were not considered to participate in the study since no sample was collected due to loss of follow up by the emerging coronavirus pandemic quarantine. Sputum samples were collected from 11 patients at the Hospital San Marcos in Chinchiná, Caldas, and cultured in Lowenstein-Jensen media for the microbiological and genomic analysis of *M. tuberculosis*. At the beginning of the COVID-19 pandemic, between March to June 2020, there were no growth-dependent or molecular susceptibility tests as part of the standard of care, therefore, no phenotypic or genotypic antimicrobial resistance data was available for the analysis. Positive cultures with colonies suspected of MTB were send to the Laboratory of Microbiology of the Universidad Católica de Manizales, Caldas for DNA extraction.

Seven samples passed the DNA quality control for the sequencing platform, whereas one sample was withdrawn because it was not taxonomically assigned to MTB genome after the bioinformatic analyzes; finally, six samples were considered for the study (Fig 1).

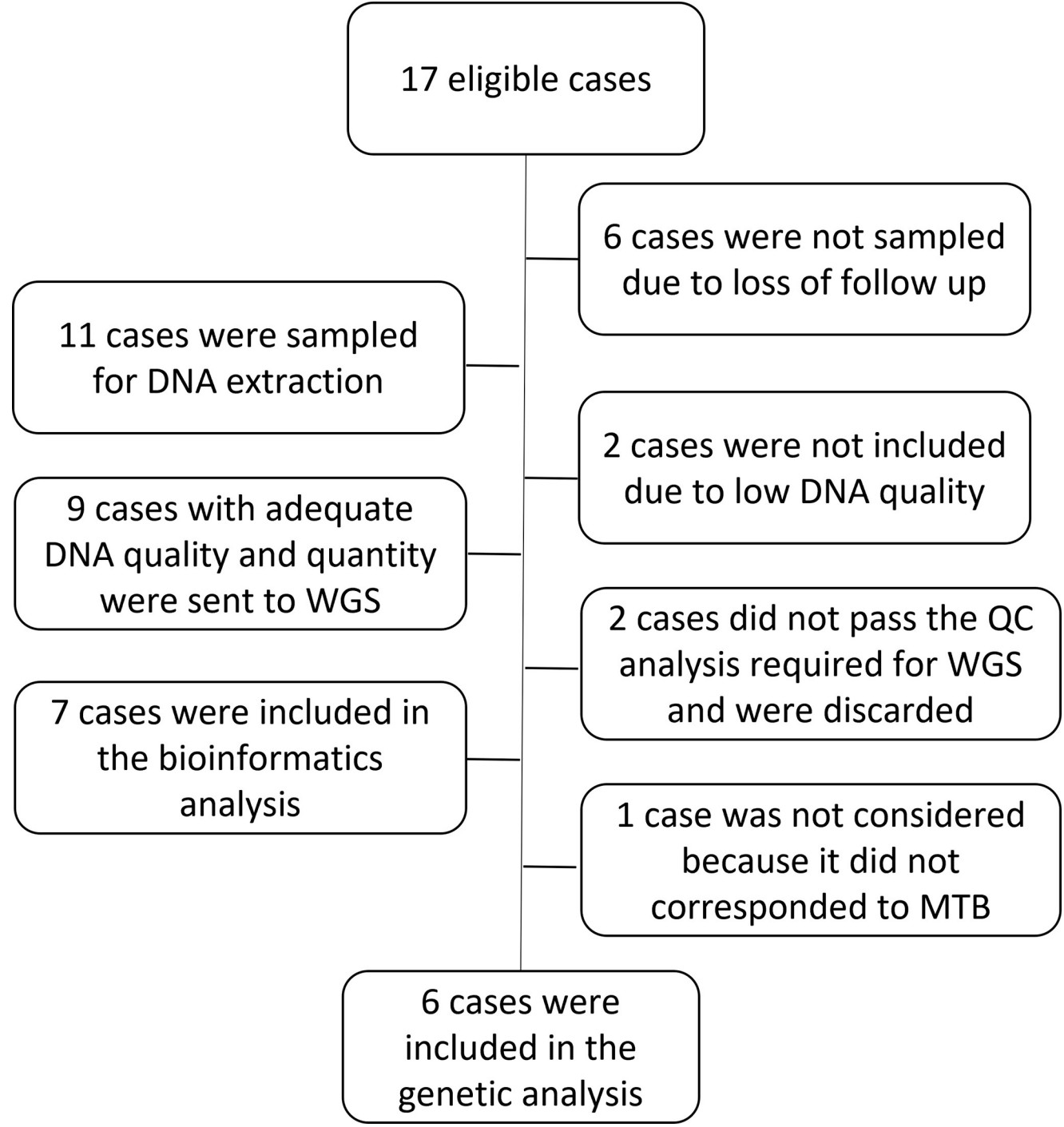

**Fig 1. Flow diagram for patient samples.**

## DNA extraction

Genomic DNA was extracted from colonies using the Bacteria DNA preparation kit following the manufacturer's protocol (Jena Bioscience, Germany). The quality and quantity of isolated DNA was measured using a Nanodrop 2000 spectrophotomer (ThermoScientific, Waltham,

MA) and a Quantus fluorometer (Promega, Madison, WI). DNA integrity was assessed by visualization in 1% agarose gel electrophoresis. DNA samples that fulfill the quality standards in terms of integrity, purity and quantity were sent to whole genome sequencing at NovoGene (Beijing, China).

## Whole genome sequencing

Sequencing was performed on PCR-free libraries on the Illumina Hiseq 2500 platform (Illumina Inc., San Diego, CA), producing 150 bp paired end reads. Quality control data of the DNA samples pre- and post-sequencing are presented on S1 and S2 Tables of the S1 File.

## Bioinformatic pipeline

The Illumina reads were analyzed in a standard PC (Intel Core i7-7700, 16 Gb of RAM, 4 TB HDD) with Linux subsystem for Windows (Ubuntu 20.04.1 LTS). An *in-house* bioinformatics workflow was installed using Miniconda (Conda 4.10.3 Python 3.9.5) with the core programs of the unified analysis variant pipeline of the Related Sequencing TB data platform (ReSeqTB/ UVP) to identify and assign lineage to the MTB isolates sequence data [4]. (S3 Table in S1 File). Briefly, reads were classified by Kraken version 2 to detect possible contamination or the presence of other mycobacteria; [5] additionally, reads were checked, validated, and trimmed with FastQC version 0.11.9 [6], FastQVAlidator (http://genome.sph.umich.edu/wiki/FastQVa-lidator), and Trimmomatic [7], respectively. Processed reads were mapped using BWA-mem [8] using H37Rv (NCBI ID: NC_000962.3) as reference genome. Alignment quality control was assesed using QuallyMap [9] and Alfred was used to evaluate the depth of coverage in all loci of interest associated with resistance lineage assignment [10]. Genomes with > 75X of depth and > 95% reference mapped with all ReSeqTB/UVP loci of interest were considered for downstream analysis. Variant calling using GATK [11] with hard-filter settings was used to eliminate artefactual false-positive variants. [12] Finally, NGSEP version 4.0 [13] and SnpEff 5.0 [14] was used to perform functional annotation of single nucleotide variants (SNVs) and insertions or deletions (INDELs). By parsing the annotated VCF it was possible to identify variants with low, medium, and high confidence (LR+>10; p<0,05) of association with antituberculous resistance and, also, identify variants not yet described in the ReSeqTB database (V. 2019-I) [15]. The pipeline for WGS TB resistance analysis is available on demand on protocols.io (https://www.protocols.io/private/A0CEADBAFBBB11EB878B0A58A9FEAC02). Links to references of the software used for the bioinformatic pipeline are summarized on S4 Table of the S1 File.

**De novo assembly and annotation.** The final high-quality reads were assembled using Megahit version 1.2.9 [16]. (S5 Table in S1 File). The assembly was carried out with a kmer range from 29 to 141 at intervals of 20 and 800 bp as minimum contig size. Kaiju version 1.7.3 [17,18] was used to perform contig taxonomic assignment. Quast version 4.3 [19] was used to generate metrics and evaluate the quality of the assemblies using *M. tuberculosis* strain HR37v (NC_000962) as reference genome. The completeness of the assembly process was verified with Busco version 4.1.4 [20]. The structural and functional annotation was performed with Prokka version 1.12–14 [21].

**Lineage assignment and phylogeny.** *M. tuberculosis* complex (MTBC) lineages/subli-neages and *in silico* spoligotyping were determined using TB-Profiler version 2.8.13 [22]. The variants on genes *rpoB, inhA, embB, embC, gyrA, gyrB, katG, panD, pncA, recA, 16S, pssC, ahpC, cycA, clpC, aftA, atpE, mas, eccC5, kasA, gpsI, mas,* and *rpsA*, were used as a second approach to classify the isolates on specific lineages and sub-lineages as was proposed by Coll et al. [23] Whole genome dinucleotide variants that passed the quality filters, including the

phylogenetically informative SNPs for MTBC typing, and excluding regions of erroneous mapping [24,25], were used to perform clustering by phylogenetic analysis using the software Mega X 10.1. Initially, a maximum likelihood (ML) analysis was run to find the best nucleotide substitution model for the dataset, the phylogenetic analysis was done by the ML method and general time reversible model with a bootstrapping test of 1000 replications, a non-rooted phylogenetic tree to show sample's clustering was obtained [26].

**Analysis of variants associated with first- and second-line drug resistance.** After variant calling and annotation, variants on each resistance gene (according to ReSeqTB/UVP) with a high confidence of association with antimicrobial resistance (LR+ > 10; p <0,05) were parsed from the variant call format (VCF) files (for a complete list of genes tested per drug, see Supplementary file, S6 Table in S1 File).

Genes related to resistance to first-line drug therapy [rifampicin (R), isoniazid (H), pyrazinamide (Z), and ethambutol (E)] and second-line drugs [fluoroquinolone (FQ), linezolid (Lnz), bedaquilina (BDQ), delamanid (DLM)] were considered for the analysis. Variants of uncertain effect in genes related to resistance to therapy, and not yet listed in the ReSeqTB database (V. 2019-I) were also evaluated. Finally, all variants (SNV, and INDELS) in drug targets were tested with PROVEAN algorithm to predict deleterious effects in proteins [27].

**Ethics statement.** The local San Marcos hospital sent all positive MTB cultures to DNA extraction after finished all diagnostic procedures during patient care. Clinical data of patients from surveillance records were collected and anonymized after patient care. All samples were deemed of retrospective nature by the clinical and academic ethics committee (San Marcos Hospital–2019–2907, and Universidad Católica de Manizales–DIP-1260-001-4184).

# Results

## Sociodemographic and clinical characteristics

All patients were of low socioeconomic status with a median age of 32 years. HIV comorbidity was present in two TB patients. Bacillary load was greater than two crosses in the first smear microscopy in four patients, however, no relation of this issue was observed with the clinical outcome (Table 1). A complete sociodemographic, epidemiological, and clinical data is listed on S7 Table in S1 File.

## Lineage assignment and phylogeny

The analysis showed that all isolates belong to the Euro-American lineage (Lineage 4), divided in LAM and Harlem sublineages (Fig 2).

In total, 1753 dinucleotide variants of high quality were found in the genome. After filtering regions of erroneous mapping, a final data set of 1386 SNVs was obtained. The un-rooted tree shows two main clades, with a branch support of 100% according to the bootstrap test.

The phylogenetic tree shows a reliable clustering. An appreciable genetic distance is not observed since there are few variations in the selected phylogeny genes.

Clade one contained samples TB02 (4.3.4.1), TB03 (4.3.4.2) and TB09 (4.3.3) and assigned to the spoligotype LAM and the closest relationship to the reference H37Rv genome. The second clade groups samples TB04, TB05, and TB06, assigned to the spoligotype Haarlem sublineage 4.1.2.1 (Fig 2). Of note, according to phylogeny analysis, the two closest isolates were TB05 and TB06 both from patients with previous antituberculous treatment.

Phylogeny inferred by maximum likelihood method using the general time reversible model, bootstrap test of 1000 replications. Values on nodes corresponds to the bootstrap value in percentage.

**Table 1. Sociodemographic and clinical characteristics of patients with tuberculosis.**

| Patient ID | TB02 | TB03 | TB04 | TB05 | TB06 | TB09 |
|---|---|---|---|---|---|---|
| *Sociodemographic* | | | | | | |
| Age | 24 | 24 | 25 | 66 | 51 | 39 |
| Sex | M | M | M | F | M | M |
| Occupation | Informal jobs | Agricultural | Agricultural | Informal jobs | Unemployed | Agricultural |
| High-risk population | No | Rural disperse | Rural disperse | No | Homeless | Rural disperse |
| *Clinical* | | | | | | |
| Comorbidities | HIV Low BMI | Low BMI | HIV Low BMI | COPD | Low BMI | None |
| Previous TB treatment | No | Yes | No | Yes | Yes | No |
| Hospitalization | Yes | No | Yes | Yes | No | Yes |
| Malnutrition | Yes | No | Yes | Yes | No | Yes |
| Weight loss | Yes | No | Yes | Yes | No | Yes |
| Hemoptisis | Yes | No | Yes | No | Yes | Yes |
| Fever | No | Yes | No | Yes | Yes | No |
| *Bacillary load* | +++ | +++ | + | ++ | ++ | + |

M: Male, F: Female, HIV: Human immunodeficiency virus, COPD: Chronic obstructive pulmonary disease, BMI: Body Mass Index.

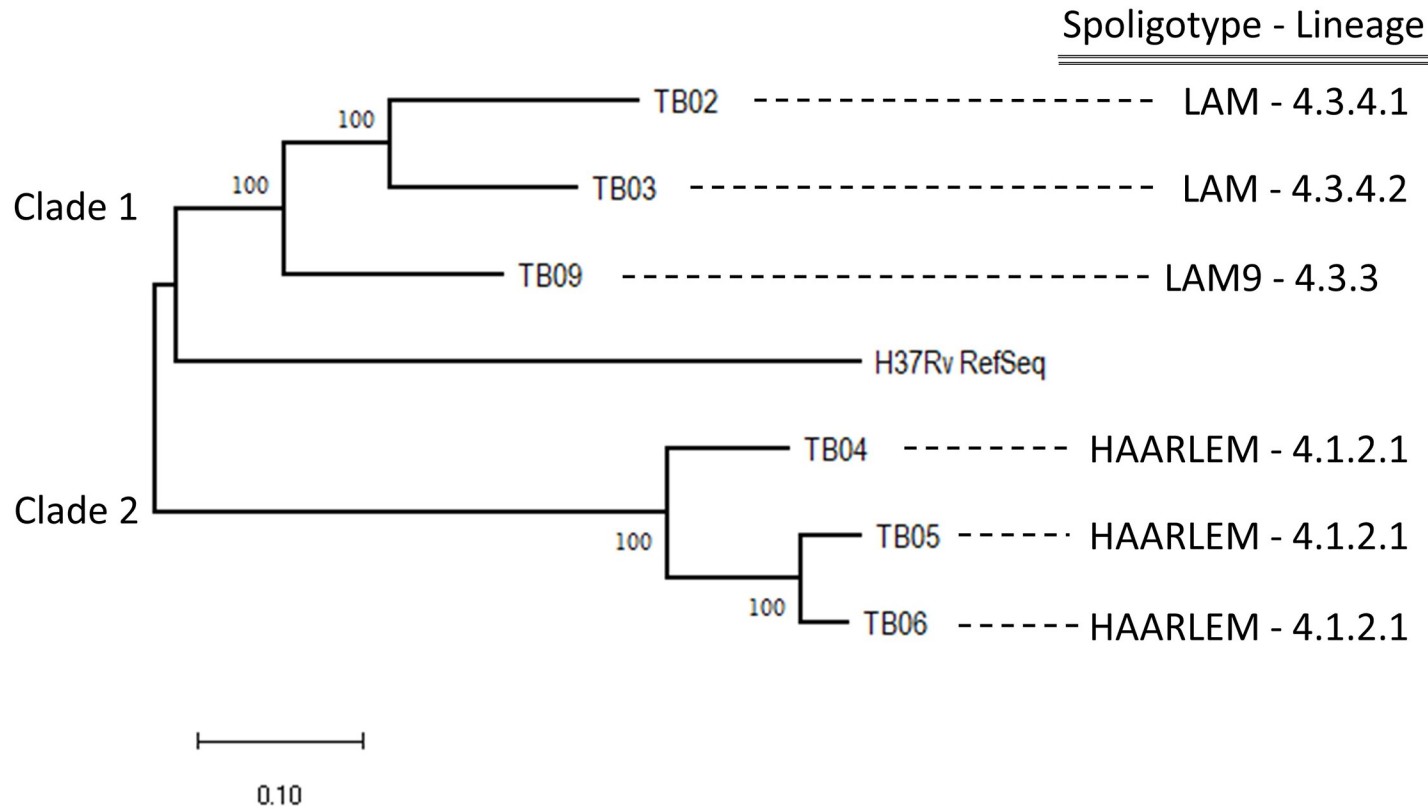

**Fig 2. Clustering by phylogenetic analysis and lineage assignment of six M. tuberculosis isolates form Chinchiná (Caldas, Colombia).**

## WGS prediction of drug resistance

According to the genomic analysis all isolates are sensitive to all the first-line drugs. Several canonical genes related to first-line treatment showed different substitutions previously reported in the ReSeqTB database with 57.8% synonymous mutations and 42.1% non-synonymous. All substitutions showed no association with clinically significant drug resistance (all positive likelihood ratio (LR)<10). A mutation in *ponA1* (1893_1894insTCG) showed LR+ >10, nevertheless, was non-significant ($p = 0,3818$). Of 54 mutations across all samples reported in ReSeqTB for first-line drugs, only three mutations (5,5%) were classified as deleterious by the PROVEAN score (S8 Table in S1 File).

Regarding second-line treatments, several mutations previously described in ReSeqTB for FQN were found. Mutations in two genes, *gyrA* (with both Glu21Gln and Gly668Asp SNV in all but one isolate) and *eccC5* (with one non-synonymous substitution in one sample). The rest of the mutational spectrum for quinolones are available in S3 Table in S1 File. Of note, one *gyrA* mutation (739G>A, Gly247Ser) with the prediction of deleterious effect (PROVEAN score of -5,942) was present in one sample. However, none of the variants related to FQN genes showed impact on drug resistance (LR+ <10, p value>0,05). Of those drugs recently added to MDR treatment, loci related to LNZ resistance (genes *rrl*, *rplC*) showed no SNV or INDEL. Another cornerstone of MDR-TB treatment, the repurposed drug BDQ, showed one synonymous substitution *atpeE* (24C>T) in one sample. Genes related to new drugs DMN and protamanid showed a single substitution in *fbiC* (Ala132Val) in one isolate, and substitution Lys270Met in *fgd1 was* found in four samples (S8 Table in S1 File). Finally, ten variants were predicted as deleterious in proteins associated with antimicrobial resistance genes. Of note, a Ala132Val mutation in *fbiC* gene related to DMN and a Thr202Ala mutation in *thyA* gene related to PAS deem deleterious PROVEAN scores with a predicted effect on the target protein which were no described in the ReSeqTB database (Table 2).

In genomes from two HIV patient isolates (TB02, TB04), the number of missense variants (541 and 533) was within range (485–546) comparing to the rest of the dataset. Both samples had 15 SNVs in genes related to first-line treatment and 10 to 17 SNVs for the second-line treatment loci, respectively (S8 Table in S1 File).

## Discussion

The present study is the first to provide information on the phylogenetic characteristics and the identification of the predominant strains in Chinchiná, a municipality with a high

**Table 2. Mutations no reported in ReseqTB in loci related to antituberculosis therapy detected by WGS analysis from isolates of Chinchiná.**

| Gene | Genome position | Mutation type | Nucleotide change | Aminoacid change | PROVEAN score | Isolates |
|---|---|---|---|---|---|---|
| **PYRAZINAMIDE (P)** | | | | | | |
| *mas* | 3282700 | missense variant | 16A>G | Thr6Ala | neutral (-1,765) | TB04 |
| **BEDAQUILINE (BDQ)** | | | | | | |
| *atpE* | 1461068 | synonymous variant | 24C>T | Gly8Gly | neutral (0) | TB03 |
| **DELAMANID (DMN)** | | | | | | |
| *fgd1* | 491591 | missense variant | 809A>T | Lys270Met | neutral (-2,158) | TB04 TB05 TB06 |
| *fbiC* | 1303325 | missense variant | 395C>T | Ala132Val | deleterious (-3,392) | TB03 |
| **PAS** | | | | | | |
| *thyA* | 3073868 | missense variant | 604A>G | Thr202Ala | deleterious (-4,577) | TB03 TB09 |

**NOTE**: PROVEAN prediction scores < -2,5 are considered deleterious.

incidence of TB; this study alsodescribes for the first time the use of WGS in the surveillance of TB drug susceptibility in Colombia. By developing a strategy of outsourcing library creation and sequencing, plus using standard computer capabilities, our team showed that it is possible to add WGS results to the surveillance data without significant changes in the laboratory setup or considerable investments in computing power. Furthermore, to our knowledge, it provides the first high-quality WGS dataset of *M. tuberculosis* from Colombia in compliance with new standards in terms of depth, coverage, tools, and quality controls during the bioinformatic pipeline (Q scores > 30, depth > 125x, MAP-Q > 58, all UVP-Loci of Interest Covered) [4,25,28]. The dataset of our study will allow the inclusion of MTB WGS from Colombia for comparative analyses using current pipelines deployed for surveillance [4,20,25,28].

Although there was a limitation on the surveillance activities due to the COVID-19 pandemic and no availability of phenotypic or molecular tests to assess drug susceptibility, the incidence of TB cases in Chinchiná for the first semester of 2020 (N = 17) was similar to reports from previous years: 2019 (N = 20), 2018 (N = 22), and 2017 (N = 19). The sample was also representative on demographic and clinical characteristics of patients, predominantly male, uneducated, with low income, and no formal jobs following local and regional patterns of mobility and social risk factors [2,29]. Furthermore, all isolates were susceptible to antimicrobial drugs according to the WGS analysis, which agrees with local epidemiology (TB-DR prevalence of 1–2%) [2,29].

The Euro-American lineage found in this study presents a variable distribution in Colombia. Similar results have been found in other regions of the world. Reports from Argentina, Brazil, México and Peru have established that the predominant strains have evolved from this lineage [30]. This lineage has been described as more transmittable than others [31]. Half of the cases analyzed corresponded to the LAM sublineage, the most frequent and globally extended of the Euro-American lineages. The other half, to the Haarlem sublineage, probably consequence of human migration and their commercial activities [32]. The European origin and the influence of European settlers in America and Colombia are confirmed. Similar results have been described by other authors in regions of the country such as Cali [33] Bogotá [34] and Medellín [35]. Contrary to these reports, in Buenaventura, a region of Valle del Cauca, the Beijing lineage, clearly associated with resistant cases has been identified [33]. Most recent reports on TB-MDR Colombian isolates using spoligotyping showed unique TB-MDR strains with no support of active circulation of MDR clones and no relation to specific lineage (Beijing vs. LAM) [36]. Moreover, there is an extensive circulation (53%) of orphan lineages between the indigenous population [36]. Both studies emphasize the importance of using WGS as the primary method for phylogenetic analysis due to high resolution and discriminatory power and potentially characterize new lineages and outbreaks [37,38].

All mutations regarding first-line drugs showed no association with resistance when using a clinical relational database for interpreting the significance of variants (10.1038/s41598-018-33731-1), even though 5,5% are predicted to show deleterious effects in the proteins using the PROVEAN score [39].

The Asp103Asp mutation is a silent mutation in the rpoB gene reported by other authors in mono-resistant isolates [40]. The Gly594Glu mutation has been considered compensatory in isolates resistant to rifampicin, although it is not found in the resistance determining region (codon 432 to 438).

Mutations in *mshA* (Ans111Ser) were detected in three isolates, although it has been reported as a variant related to the Haarlem lineage and responsible for conferring resistance to Ethionamide [41,42] there is no significant association with drug resistance in the ReSeqTB database [4].

Regarding BDQ resistance by *atpeE* mutations, one isolate showed a non-repoted synonymous mutation (24C>T) outside codon positions 53 and 72, the key genomic region with most substitutions related to an increase in MIC for BDQ [43]. DMN, and more recently pretomanid for TB-XDR treatment, use similar pathways and genes for activation. *fgd1* encodes a key enzyme for the recycle of the target $F_{420}$ of DMN; in our study, we found three isolates with Lys270Met substitutions in patients not previously exposed to the drug, this substitution was phylogenetically restricted to the Harleem lineage which is regarded as neutral [43]. *fbiC* encodes a protein that participates in $F_{420}$ biosynthesis and harbors key mutations found in resistant MTB isolates. The mutation Ala132Val we detected in one isolate was not described in the systematic review on DMN resistance by Kadura *et al.*, [43] however, it showed a deleterious effect on the protein according to the PROVEAN score (-3,392); thus, the presence of this mutation in TB isolates not previously exposed to the drug requires future surveillance as not seems to be lineage-restricted [44].

No differences in SNV and other variants were observed when comparing patients living with HIV within our sample. Although with did not find statistical significance due to the small sample, our results are broadly consistent with the literature. Previous reports found that HIV co-infection did not affect the mutation rate and was not associated with faster *de novo* development of drug resistance [45,46].

As for the patients with previous treatment, our results showed full genetic susceptibility to current antimicrobial drugs and successful clinical outcomes after the new HRZE treatment. Although there was no previous microbial culture in our samples, the WGS results in the context of adequate therapeutic adherence and no cavitation with fibrosis in radiologic studies (associated with poor tissue penetration of drugs) [47] seem to indicate that a re-infection with a drug sensitive strain by re-exposure to a known focus would be the most plausible explanation for the re-treatment cases.

There were some limitations in our study, since due to the COVID pandemic, the researchers did not have access to phenotypic or molecular tests that would have made it possible to compare data with our genomic findings; and, due to international shipping for WGS and the corresponding response times, the delivery of the results added approximately four weeks to analyze them. Currently, we are improving those aspects for future WGS analyzes of MTB surveillance projects.

## Conclusions

Recent studies in top medical journals have shown the enormous potential of WGS in the surveillance of tuberculosis but at the cost of high investments in laboratory infrastructure, sequence capacity, and processing power. However, our results indicate that it is possible to develop genomics surveillance applications at the regional level without considerable investments in sequencing platforms, laboratory upgrades, or computational infrastructure. A decentralized strategy of drug susceptibility analysis with WGS of first and second-line anti-TB drugs could accelerate the adoption of these methodologies in developing regions. The future of this approach depends on a collaboration of multiple institutions, both public and private, to overcome the limitations in turnaround time, number of samples, large phylogenetic analysis, computational requirements, and rigorous validation processes required to include WGS data in the surveillance arsenal against TB.

## Supporting information

**S1 File.**
(XLSX)

## Acknowledgments

We thank the health professionals of the Hospital San Marcos of Chinchiná who provided insight and expertise that greatly assisted the research, and the epidemiological support obtained from the Dirección Territorial de Salud de Caldas.

## Author Contributions

**Conceptualization:** Lusayda Sánchez-Corrales, Olga Lucía Tovar-Aguirre, Narmer Fernando Galeano-Vanegas, Johan Sebastián Hernández-Botero, Fernando Siller-López.

**Data curation:** Narmer Fernando Galeano-Vanegas, Carlos Ernesto Maldonado-Londoño, Johan Sebastián Hernández-Botero.

**Formal analysis:** Lusayda Sánchez-Corrales, Olga Lucía Tovar-Aguirre, Narmer Fernando Galeano-Vanegas, Paula Alejandra Castaño Jiménez, Carlos Ernesto Maldonado-Londoño, Johan Sebastián Hernández-Botero.

**Funding acquisition:** Ruth Arali Martínez-Vega, Johan Sebastián Hernández-Botero.

**Investigation:** Lusayda Sánchez-Corrales, Paula Alejandra Castaño Jiménez, Fernando Siller-López.

**Methodology:** Narmer Fernando Galeano-Vanegas, Paula Alejandra Castaño Jiménez.

**Project administration:** Lusayda Sánchez-Corrales, Olga Lucía Tovar-Aguirre, Fernando Siller-López.

**Resources:** Ruth Arali Martínez-Vega.

**Software:** Narmer Fernando Galeano-Vanegas, Carlos Ernesto Maldonado-Londoño, Johan Sebastián Hernández-Botero.

**Supervision:** Fernando Siller-López.

**Validation:** Ruth Arali Martínez-Vega, Johan Sebastián Hernández-Botero.

**Writing – original draft:** Lusayda Sánchez-Corrales, Olga Lucía Tovar-Aguirre.

**Writing – review & editing:** Narmer Fernando Galeano-Vanegas, Johan Sebastián Hernández-Botero, Fernando Siller-López.

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
