## [Decision Letter · Decision Letter 0]

7 Jul 2021

PONE-D-21-17481

Phylogenomic analysis and Mycobacterium tuberculosis antibiotic resistance prediction by whole-genome sequencing from clinical isolates of Caldas, Colombia.

PLOS ONE

Dear Dr. Siller-Lopez, 

Thank you for submitting your manuscript to PLOS ONE. After careful consideration, we feel that it has merit but does not fully meet PLOS ONE’s publication criteria as it currently stands. Therefore, we invite you to submit a revised version of the manuscript that addresses the points raised during the review process.

We look forward to receiving your revised manuscript.

Kind regards,

Seyed Ehtesham Hasnain

Academic Editor

PLOS ONE

Journal Requirements:

2. Please provide additional details regarding participant consent. In the ethics statement in the Methods and online submission information, please ensure that you have specified (a) whether consent was informed and (b) what type you obtained (for instance, written or verbal, and if verbal, how it was documented and witnessed). If your study included minors, state whether you obtained consent from parents or guardians. If the need for consent was waived by the ethics committee, please include this information.

For additional information about PLOS ONE ethical requirements for human subjects research, please refer to " ext-link-type="uri" xlink:type="simple">http://journals.plos.org/plosone/s/submission-guidelines#loc-human-subjects-research."

"This research was supported by Universidad Católica de Manizales, Universidad de Manizales and Universidad de Santander. Epidemiological support was obtained from the Dirección Territorial de Salud de Caldas."

"The funders had no role in study design, data collection and analysis, decision to

publish, or preparation of the manuscript."

4. We note that Figure 3 in your submission contain satellite images which may be copyrighted. All PLOS content is published under the Creative Commons Attribution License (CC BY 4.0), which means that the manuscript, images, and Supporting Information files will be freely available online, and any third party is permitted to access, download, copy, distribute, and use these materials in any way, even commercially, with proper attribution. For these reasons, we cannot publish previously copyrighted maps or satellite images created using proprietary data, such as Google software (Google Maps, Street View, and Earth). For more information, see our copyright guidelines: http://journals.plos.org/plosone/s/licenses-and-copyright.

a) You may seek permission from the original copyright holder of Figure(s) [#] to publish the content specifically under the CC BY 4.0 license.

Additional Editor Comments (if provided):

Major Revision

Reviewers' comments:

Reviewer's Responses to Questions

**Comments to the Author**

1. Is the manuscript technically sound, and do the data support the conclusions?

Reviewer #1: Yes

Reviewer #2: Yes

2. Has the statistical analysis been performed appropriately and rigorously? 

Reviewer #1: N/A

Reviewer #2: Yes

3. Have the authors made all data underlying the findings in their manuscript fully available?

Reviewer #1: Yes

Reviewer #2: Yes

4. Is the manuscript presented in an intelligible fashion and written in standard English?

Reviewer #1: Yes

Reviewer #2: Yes

5. Review Comments to the Author

Reviewer #1: The authors have carried out work entitled “Phylogenomic analysis and Mycobacterium tuberculosis antibiotic resistance prediction by whole-genome sequencing from clinical isolates of Caldas, Colombia.” The study has explored the variants present in MTB clinical samples from

Chinchiná, Caldas, a small town in Colombia with a high incidence of TB.

However, there is some gap in the experimental work to reach some significant outcome which authors need to address.

Reviewer #2: Comments:

Present study has been conducted by Sánchez-Corrales et al. and trying to represent the molecular characterization as well as genomic surveillance of MTB strains from Chinchiná, Caldas. WGS would be the best way to represent genomic surveillance. However, if we talking about WGS in routine process, it would be difficult for middle and low income countries. Following are some suggestions which can make the manuscript stronger and beneficial for the readers:

Introduction:

• Write Mycobacterium tuberculosis (M. tuberculosis). (Line no. 45)

• Required reference (agent before the arrival of SARS-CoV-2….) (Line no. 49)

• Modify the data from Global TB report 2020.

Method:

• Ethical statement should write in Method section.

Results:

• It is advised to take permission from source (Google) for Geographical territorial picture for any international publication. This may be a copyright issue for future (Figure 3).

• While analyzing the sequence data, it is advised to include or discuss about the resistance patterns for newer drugs [beqaquiline and delamanid] as well as repurposed drugs [clofazimine].

6. PLOS authors have the option to publish the peer review history of their article (what does this mean?). If published, this will include your full peer review and any attached files.

Reviewer #1: No

Reviewer #2: No

---

## [Author Response · Author response to Decision Letter 0]

10 Sep 2021

Journal Requirements:

Thank you, we have corrected the style of our manuscript according to the PLOS ONE requirements.

2. Please provide additional details regarding participant consent. In the ethics statement in the Methods and online submission information, please ensure that you have specified (a) whether consent was informed and (b) what type you obtained (for instance, written or verbal, and if verbal, how it was documented and witnessed). 

… If your study included minors, state whether you obtained consent from parents or guardians. If the need for consent was waived by the ethics committee, please include this information.

We have now specified the ethics statement in the Methods and online submission information. According to the type of study (a retrospective study of medical records) we did not require an informed consent. Data was totally anonymized and discussed and approved by two ethical committees as it is now stated at the ethical statement section. 

… If you are reporting a retrospective study of medical records or archived samples, please ensure that you have discussed whether all data were fully anonymized before you accessed them and/or whether the IRB or ethics committee waived the requirement for informed consent. If patients provided informed written consent to have data from their medical records used in research, please include this information.

N.A.

… Once you have amended this/these statement(s) in the Methods section of the manuscript, please add the same text to the “Ethics Statement” field of the submission form (via “Edit Submission”).

Done

3. … We note that you have provided funding information that is not currently declared in your Funding Statement. However, funding information should not appear in the Acknowledgments section or other areas of your manuscript. We will only publish funding information present in the Funding Statement section of the online submission form. Please remove any funding-related text from the manuscript and let us know how you would like to update your Funding Statement.

We amendment the Acknowledgment section and remove the funding-related text from the manuscript. The funding statement continues to be as presented: "The funders had no role in study design, data collection and analysis, decision to publish, or preparation of the manuscript."

4. We note that Figure 3 in your submission contain satellite images which may be copyrighted. All PLOS content is published under the Creative Commons Attribution License (CC BY 4.0), … We require you to either (a) present written permission from the copyright holder to publish these figures specifically under the CC BY 4.0 license, or (b) remove the figures from your submission:

Thank you for your indication. We deleted Figure 3 after reviewing that no major contribution to the manuscript was obtained from that figure.

Thank you for your observation. We now presented the ethics statement only in the Methods section of our manuscript. 

 

Detailed responses to reviewers:

Reviewer # 1

Major comments:

1. For the prediction of drug resistance causing variants against bedaquiline and delamanid drugs which are recently FDA approved drugs for the treatment of TB, the authors can consider recently published articles related to these drugs. They can report whether variants related to these drugs are present in the analyzed clinical samples.

Thank you for this direction. Text has been revised to incorporate the reviewer comment. We modified Table 2 and S8 Table to include information on the nucleotide sequence of the TB isolates of our study related to variants associated to bedaquiline and delamanid resistance. Additionally, it was included this sentence in results section; “Another cornerstone of MDR-TB treatment, the repurposed drug BDQ, showed one synonymous substitution atpeE (24CT) in one sample. Genes related to new drugs DMN and protamanid showed a single substitution in fbiC (Ala132Val) in one isolate, and substitution Lys270Met in fgd1 was found in four samples (S8 Table)”; and “Finally, ten variants were predicted as deleterious in proteins associated with antimicrobial resistance genes. Of note, a Ala132Val mutation in fbiC gene related to DMN and a Thr202Ala mutation in thyA gene related to PAS deem deleterious PROVEAN scores with a predicted effect on the target protein which were no described in the ReSeqTB database”. Later we discussed in the next section about findings from line 293 to 305.

2. The authors have reported that their samples of HIV+TB (Co-morbidity) and TB patients. However, they have analyzed the samples in one group only. It will add significanse value to the study if they compare the variants of TB patients with patients having comorbdity condition.

It was added to the text, to include some about it in lines 231 to 234, discussed at lines 308 to 312

3. Authors are giving synonymous variants. They should explain why and how does it affect the drug resistance development on MTB?

Reviewer is right In silico we have not evidence about it. However, we consider it is important that data remains in order to see that there is no evidence trying to correlate the trait with the variant. We know that it would sound obvious but we tried to avoid commitments with any results in this case because changes due to this kind of mutations are more related to transcription regulations, accordingly to Pevsner in an eukaryotic model. For that reason, we include the sentence “the researchers did not have access to phenotypic or molecular tests that would have made it possible to compare data with our genomic findings” 320-321.

4. Authors have mentioned, “Likewise, for further analysis it would be considered to carry out protein modeling to determine if changes in the mutated amino acids could generate resistance.” However, the effect of variants on protein function can be analysed by SIFT (whether an amino acid substitution affects protein function), PROVEAN (an amino acid substitution or indel has an impact on the biological function of a protein) or PolyPhen. Authors can add this step to their bioinformatic analysis.

Thank you so much for your recommendation. We run Provean and give us some interesting results discussed in the document’s body.

Minor comments:

1. Abstract

5. “All samples were proven sensitive to antituberculosis drugs by phenotype and genomic analysis, although unreported mutations were identified.” Authors have mentioned phenotype, however they have not carried out Drug susceptibility testing, therefore this sentence should be modified, Line nos. 36-37; Page No. 2.

Thank you for the observation. Although we did not perform the drug susceptibility tests by ourselves, we did receive the drug susceptibility results from the Colombian National Health Institute where a mandatory replicate sample is sent to report the clinical case. We have now included that statement on the corresponding Materials and methods section. 

2. Introduction

6. Should be read as, “1.7 million” for “1,7 million”, Line no. 47; Page No. 3.

7. Should be read as, “100,000 habitants at” for “100.000 habitants at” Line no. 52; Page No. 3.

8. “Also, outbreaks can be identified with higher resolution, allowing optimized control measures not only by revealing the genetic makeup and evolution of strains, but by showing epidemiological evidence of contagion from patients unwilling or unable to give information for the study of the case, as its routinely the case for high-risk social groups like homeless, immigrants, and itinerant agricultural workers.” Sentence is too long. Rewrite it. Line no. 66; Page No. 3: Line nos. 67-70; Page No. 4.

9. Should be read as, “visualization in” for “visualization by”, Line no. 96; Page No. 5.

10. 1. The authors have repeated this section from “The readings were reviewed [4], validated (FastQVAlidator v1.0.5 2017) and trimmed (Trimmomatic v0.36) and the processed reads were mapped using BWA v0.7.17 [19] with the M. tuberculosis reference genome H37Rv (NC_000962.3). Quality control of the alignment was performed using QuallyMap v2.2.1 [6] for coverage statistics and Phred score to assess depth and coverage at all loci of interest associated with resistance. Only genomes 75X deep and 95% reference mapped to all UVP v2.7.0 loci of covered interest were considered for further analysis. Variant calling was accomplished with the use of GATK v4.2.0.0 [20] with filter adjustments to eliminate artificial false positive variants. Finally, NGSEP v4.0 was used to score both SNV and INDEL. [8] When analyzing the annotated VCF file, it was possible to identify variants with low, medium, and high confidence (LR +10; p 0.05)” from De novo assembly and annotation section. The authors can write, “The variants obtained after variant calling were considered for the prediction of drug reistance” Line nos. 144-154; Page No. 7.

11. Asn927Asn is not non-synonymous mutation. Please check it, Line nos. 223; Page No. 11. 

12. In Table 5. Variants related to drug resistance are listed. Out of 19, 42% variants are synonymous variants which should be excluded from the study as they will have no effect on the functional aspect of the gene. 

13. Should be read as, “these samples were” for “this sample was”, Line no. 257; Page No. 15.

14. “Other authors evaluated the variation that the protein present in these mutations could have and found that the effect of the mutation on it was neutral.” Rewrite this sentence. Line nos. 296-298; Page No. 16.

15. “The Asp103Asp mutation is a silent mutation in the rpoB gene reported by other authors in mono-resistant isolates”, Do this mutation affect the physiology of MTB? Line nos. 299-300; Page No. 17.

16. Should be read as, “in such a way this activity” for “in such a way that this activity” Line no. 323; Page No. 18.

17. Should be read as, “that allow to identify local” for “that allow identifying local” Line no. 333; Page No. 18.

18. 1. Should be read as, “use of WGS for the characterization” for “use of the WGS the characterization” Line no. 337; Page No. 18.

19. Figure. 1: In the flow chart, in last box please check spelling of “analyses”.

20. “Table 2. Clinical characteristics of pulmonary tuberculosis patients from Chinchiná, Colombia.” is not present in the manuscript, Line no. 191; Page No. 9.

We thank the reviewer for highlighting all these points. We have amended the manuscript considering the observations. Since the restructuring of the manuscript was substantial, we have incorporated the reviewer suggestions as part of the major changes of the manuscript.

---

## [Editor Report · Decision Letter 1]

27 Sep 2021

Phylogenomic analysis and Mycobacterium tuberculosis antibiotic resistance prediction by whole-genome sequencing from clinical isolates of Caldas, Colombia.

PONE-D-21-17481R1

Dear Dr. Siller-Lopez,

We’re pleased to inform you that your manuscript has been judged scientifically suitable for publication and will be formally accepted for publication once it meets all outstanding technical requirements.

Kind regards,

Seyed Ehtesham Hasnain

Academic Editor

PLOS ONE

Additional Editor Comments (optional):

I have gone through the revised manuscript and also the Author's response to the comments of the reviewers. The important issue of synonyms variants and drug resistance has been addressed. They have also added results of PROVEAN analysis. In my view, Authors have comprehensively revised the manuscript addressing all the comments of the reviewers. All typo and grammatical errors have been taken care off. All other explanations provided by the Authors to the queries of the reviewers are quite satisfactory.

I recommend this manuscript for publication.
---

## [Editor Report · Acceptance letter]

29 Sep 2021

PONE-D-21-17481R1 

Phylogenomic analysis and *Mycobacterium tuberculosis* antibiotic resistance prediction by whole-genome sequencing from clinical isolates of Caldas, Colombia 

Dear Dr. Siller-López:

I'm pleased to inform you that your manuscript has been deemed suitable for publication in PLOS ONE. Congratulations! Your manuscript is now with our production department. 

Kind regards, 

on behalf of

Prof. Seyed Ehtesham Hasnain 

Academic Editor

PLOS ONE